# A QbD Approach to Design and to Optimize the Self-Emulsifying Resveratrol–Phospholipid Complex to Enhance Drug Bioavailability through Lymphatic Transport

**DOI:** 10.3390/polym14153220

**Published:** 2022-08-08

**Authors:** Syed Abul Layes Gausuzzaman, Mithun Saha, Shahid Jaman Dip, Shaiful Alam, Arup Kumar, Harinarayan Das, Shazid Md. Sharker, Md Abdur Rashid, Mohsin Kazi, Hasan Mahmud Reza

**Affiliations:** 1Department of Pharmaceutical Sciences, North South University, Dhaka 1229, Bangladesh; 2Materials Science Division, Atomic Energy Centre, 4 Kazi Nazrul Islam Avenue, Shahbagh, Dhaka 1000, Bangladesh; 3Department of Pharmaceutics, College of Pharmacy, King Khalid University, Guraiger, Abha 62529, Saudi Arabia; 4Pharmacy Discipline, Faculty of Health, School of Clinical Sciences, Queensland University of Technology, Brisbane, QLD 4000, Australia; 5Department of Pharmaceutics, College of Pharmacy, King Saud University, P.O. Box 2457, Riyadh 11451, Saudi Arabia

**Keywords:** self-emulsifying drug delivery system, resveratrol–phospholipid complex, quality by design, chylomicron flow blocking approach, lymphatic drug transport, bioavailability

## Abstract

**Highlights:**

**Abstract:**

Objectives: Despite having profound therapeutic value, the clinical application of resveratrol is restrained due to its <1% bioavailability, arising from the extensive fast-pass effect along with enterohepatic recirculation. This study aimed to develop a self-emulsifying formulation capable of increasing the bioavailability of resveratrol via lymphatic transport. Methods: The resveratrol–phospholipid complex (RPC) was formed by the solvent evaporation method and characterized by FTIR, DSC, and XRD analyses. The RPC-loaded self-emulsifying drug delivery system (SEDDS) was designed, developed, and optimized using the QbD approach with an emphasis on resveratrol transport through the intestinal lymphatic pathway. The in vivo pharmacokinetic study was investigated in male Wister Albino rats. Results: The FTIR, DSC, and XRD analyses confirmed the RPC formation. The obtained design space provided robustness of prediction within the 95% prediction interval to meet the CQA specifications. An optimal formulation (desirability value of 7.24) provided Grade-A self-emulsion and exhibited a 48-fold bioavailability enhancement compared to the pure resveratrol. The cycloheximide-induced chylomicron flow blocking approach demonstrated that 91.14% of the systemically available resveratrol was transported through the intestinal lymphatic route. Conclusions: This study suggests that an optimal self-emulsifying formulation can significantly increase the bioavailability of resveratrol through lymphatic transport to achieve the desired pharmacological effects.

## 1. Introduction

*trans*-Resveratrol, a stilbenoid polyphenol and BCS Class II compound, exhibits diverse therapeutic functions, including anti-inflammatory, antioxidant, anti-diabetes, neuroprotective, cardioprotective, and anti-cancer effects. Many preclinical studies have also demonstrated resveratrol as a chemopreventive or chemotherapeutic agent [1]. However, low solubility, limited stability, and short half-life limit its effectiveness. Furthermore, resveratrol undergoes extensive fast-pass metabolism and enterohepatic recirculation, producing glucuronides and sulfates metabolites, resulting in less than 1% oral bioavailability [2]. In consequence, the promising therapeutic effects of resveratrol are compromised and reduced manifold, thus failing in human clinical trials [3]. A higher dose of resveratrol was employed to improve systemic availability and obtain the desired therapeutic effects. This approach observed minor benefits; instead, several toxicities were reported due to high doses during the clinical studies [4]. The failure to achieve adequate bioavailability at tolerable doses impedes translating preclinical pharmacological effects to equivalent efficacy in human clinical trials [5].

Rather than increasing the dose, formulation strategies have been investigated to circumvent the bioavailability hurdles of resveratrol. Due to the physicochemical limitations of resveratrol, a vast number of formulation systems were investigated, including cyclodextrin complexation, microparticle formation, nanosponges, liposphere, nanoemulsion, nano-capsule, solid lipid nanoparticles, liposome, and other systems [6]. Despite being an old drug delivery system, SEDDS was reintroduced to the oral delivery of notorious drug candidates due to its high payload capacity and high mucosal permeability and absorption capacity, circumventing first-pass metabolism and therefore improving bioavailability [7,8]. Consequently, SEDDS has been used to improve the biopharmaceutical performance of BCS Class II and Class IV drugs, oral peptides, and hydrophilic drugs utilizing the hydrophobic ion-pairing technique. Furthermore, SEDDS has been highly recommended for the oral delivery of BDDCS Class II drugs, which exhibited low solubility and extensive first-pass metabolism as per the Biopharmaceutics Drug Disposition Classification System [9,10] and investigated to improve oral bioavailability [11] and to reduce toxicological properties [12]. Like other BDDCS Class II drugs, resveratrol has low solubility and an extensive first-pass metabolism profile, making it an excellent candidate for SEDDS to resolve the bioavailability obstacle. Utilizing SEDDS as the carrier, several attempts have been carried out to improve the bioavailability of resveratrol [13,14,15,16,17,18,19]. Despite achieving the nano-sized globule, excellent release profile, high intestinal permeability, and improved absorption, the extent of bioavailability enhancements observed was rather marginal. It demonstrated that, even after a significantly increased absorption, the bioavailability of resveratrol is mainly limited by the extensive fast-pass metabolism along with enterohepatic recirculation. Therefore, bypassing fast-pass metabolism and enterohepatic recirculation should be the main objectives to enhance the bioavailability of resveratrol, which was absent in those studies.

Drug transport from the intestine to systemic circulation through the lymphatic pathway avoids hepatic fast-pass metabolism and enterohepatic recirculation. A drug to be transported through the intestinal lymphatic route should have a logP > 5 and LCT solubility of more than 50 mg/mL [20]. Even though resveratrol has a log P of 3.1 and a low LCT solubility, lymphatic transport can still be accomplished when increasing the drug’s lipophilicity [21] or formulating with long-chain lipid-based or lymphatic transport-facilitating excipients [22]. The complexation of polyphenolic compounds with phospholipids improves lipophilicity, and resveratrol was likewise complexed with phospholipids and subsequently incorporated into SEDDS. Phospholipids are also a safe biomaterial and excipient for human use compared to other lymphatic transport facilitating excipients [23]. Therefore, phosphatidylcholine was selected to form the complex with resveratrol. However, bioavailability enhancement was moderate due to the lack of lymphatic transport-facilitating excipients in SEDDS [13]. Therefore, we screened the oil phase from long-chain fatty acids, glycerides, or their synthetic derivatives, since the long-chain oil phase promotes drug absorption through the lymphatic pathway. Vitamin E TPGS (TPGS) is a biomaterial that has achieved Generally Recognized as Safe (GRAS) standard from USFDA. It exhibits solubilization enhancing and emulsifying properties. Furthermore, it increases the permeation and absorption of drugs by inhibiting P-glycoprotein (P-gp) and thus enhances the bioavailability of many poorly bioavailable drugs [24]. TPGS has also been found to increase bioavailability by increasing drug transport through the intestinal lymphatic pathway [25].

By complexing resveratrol with phospholipid and eventually loading it into SEDDS designed with lymphatic-transport facilitating excipients, we aimed to increase resveratrol’s bioavailability via lymphatic drug transport. Since SEDDS comprises a multi-excipients system and our target was to obtain the nano-sized self-formed globule, the quality characteristics could be affected by the selection of excipients, their concentrations, process parameters, and other factors [26,27]. The formulation development process of such a complex system should follow the QbD approach, as it is a holistic, risk-based, scientific, and predefined target-oriented developmental framework [28]. Therefore, the targets of Quality Target Product Profiles (QTPPs) and Critical Quality Attributes (CQAs), risk assessment, and optimization were examined while maintaining the conditions favorable to promoting the lymphatic transport of resveratrol and thereby increasing the bioavailability.

## 2. Materials

The trans-Resveratrol used in this study was purchased from Xi’an Wanfun Biotech Co., Ltd. (Xi’an, China). Labrafil^®^ M 1944 CS, Labrafil^®^ M 2125 CS, Gelucire^®^ 44/14, Gelucire^®^ 48/16, Gelucire^®^ 50/13, Labrasol^®^ ALF, Maisine^®^ CC, Transcutol^®^ HP, Plurol^®^ Oleique CC 497, and Peceol™ were kindly donated by Gattefossé (Lyon, France). Cycloheximide, Oleic acid, Propylene Glycol, Tween 20, Tween 80, Kolliphor^®^ EL, Kolliphor^®^ RH 40, n-Octanol, Acetone (HPLC grade), Ethyl acetate (HPLC grade), Ethanol (LC grade), acetonitrile (HPLC grade), and Methanol (HPLC grade) were purchased from Sigma-Aldrich Chemie GmbH (Taufkirchen, Germany) in required amounts. Lipoid S100 (phosphatidylcholine) was obtained as a generous gift from Lipoid GmbH (Ludwigshafen, Germany). Vitamin E TPGS was kindly donated by Antares Health Products, Inc. (Jonesborough, TN, USA). All other materials used in this study were of analytical grade unless otherwise stated.

## 3. Methods

### 3.1. Preparation and Characterization of the Complex

The RPC was synthesized according to the procedure described in the published patent [29]. Resveratrol and phospholipids were weighed in stoichiometric ratios of 1:1.2, dissolved in 20 mL acetone and refluxed at 50 °C for 3 h with constant stirring. The acetone was then evaporated, and the mixture cooled. Then, n-hexane was added, stirred with a glass rod, and kept overnight. Then, excess n-hexane was decanted, and a pure, dry RPC was weighed and collected. The obtained resveratrol–phospholipid complex was characterized along with resveratrol, phosphatidylcholine, and their physical mixture using Fourier-transform infrared spectroscopy (FTIR) and X-ray Diffraction (XRD).

#### 3.1.1. Characterization Using FTIR, XRD, Differential Scanning Calorimetry (DSC), and Scanning Electron Microscopy (SEM)

ATR-FTIR spectroscopy was used to determine the complexation (PerkinElmer, Model No. 65 FT-IR, Waltham, MA, USA). Approximately 50 mg of each sample of resveratrol, phospholipid, and the physical mixture of resveratrol and phospholipid (1:1.2 stoichiometric ratio), and RPC were observed directly using the spectrometer. The physical state of the pure resveratrol, phospholipid, the physical mixture of resveratrol and phospholipid, and RPC were analyzed by X-ray diffractometer (XRD) (PW 3040-X’Pert PRO PANalytical, Philips, The Netherlands) at 40 kV and 30 mA. The 2θ angle was ranged from 5–60°, using Cu Kα radiation wavelengths of 1.54059 Å with a scan step size of 0.0167°. The thermal behavior of pure resveratrol, phosphatidylcholine, the physical mixture of resveratrol and phospholipid, and RPC were performed using DSC (Perkin–Elmer, Waltham, MA, USA) within a temperature range of 25 to 350 °C. The morphology of the resveratrol–phospholipid complex was observed using Scanning Electron Microscope (Hitachi, Germany), operating at 30 kV accelerating voltage.

#### 3.1.2. Solubility Determination

An excess amount of resveratrol and RPC was added to 10 mL of water or n-octanol in a closed glass vial. The glass vials were shaken at 37 °C for 48 h, followed by centrifugation at 10,000 rpm for 10 min. The supernatant was collected and passed through a 0.2 µm syringe filter. After appropriate dilution with methanol, the concentration of resveratrol was determined by the HPLC method described in this article.

### 3.2. QbD-Based Formulation-Development and Optimization Process

#### 3.2.1. Defining QTPPs and QAs

Quality target product profiles (QTPPs) and quality attributes (QAs) were defined along with appropriate targets and justifications based on previous experience, experimental trials, and analysis of former publications. Based on the patient-centric efficacy, safety, quality, and desirability of the drug product, the impact of QTPPs was categorized as high (H), medium (M), or low (L).

#### 3.2.2. Risk Assessment (RA) Study

As part of the initial risk assessment, a diagram was generated by incorporating probable factors affecting QAs. Among them, crucial material variables and process parameters were selected for the risk assessment matrix (RAM) study using Lean QbD™ software (QbD Works LLC, Fremont, CA, USA). The risk-level of MAs and PPs were categorized as high-risk = 9, medium-risk = 3, and low-risk = 1 based on their potential to affect QAs. The RAM for QTPP-QA and QA-MA/PP were obtained as the function of their interdependent relationship. The final risk score of each QA, MA, and PP from those RAM. After the risk reduction by adapting the following experimental approaches, and further risk analysis was performed using RAM of the QA-MA/PP relationship.

#### 3.2.3. Risk Reduction by Experimental Approaches

The risk scores of some MAs/PPs were reduced by selecting the specific materials or their concentration range determination. After the risk reduction adapting following experimental approaches, the potentiality of each MAs/PPs to affect each QAs was further analyzed by RAM. The final risk score of each MAs/PPs was obtained after experimental risk reduction.

##### Screening of Oil

Six different long-chain triglycerides (LCTs) and long-chain fatty acids (LCFAs) were primarily chosen based on their potential to carry out lymphatic transport. Among the six different oils were Labrafil^®^ M1944CS, Labrafil^®^ M 2125 CS, Maisine^®^ CC, Plurol^®^ Oleique CC 497, Peceol™, and Oleic acid. Studies were carried out to screen out the oil in which the RPC showed the highest degree of solubility. RPC equivalent to 50 mg resveratrol was dissolved in 300 mg of each oil in separate vials and allowed to shake in a thermostatically controlled digital shaker for 72 h at 37 ± 0.5 °C. Then, these oils were centrifuged in a microcentrifuge device (Hanil Scientific Co., Model No. Smart R15, Gimpo, Korea) at 10,000 rpm for 15 min at room temperature. The solubility was determined by UV spectroscopy with the appropriate dilution of methanol (Shimadzu Corporation, Model No. UV-VIS 1600, Kyoto, Japan) at 306 nm wavelength.

##### Screening of Surfactant and Co-Surfactant

The selection of surfactants was carried out based on ET and TP. Eight different surfactants such as Gelucire^®^ 44/14, Gelucire^®^ 48/16, Gelucire^®^ 50/13, Labrasol^®^ ALF, Tween 20, Tween 80, Kolliphor^®^ EL, and Kolliphor^®^ RH40 were primarily chosen as potential candidates for the study. An RPC complex equivalent to 50 mg resveratrol was dissolved in 300 mg of the selected oil in separate vials, then 200 mg of each surfactant was added to each vial before dispersing them completely. From the eight samples, 500 mg of each were weighed, and using a USP Dissolution Apparatus II (Copley Scientific, Model No. NE4-COPD, Nottingham, UK); the emulsification times were observed for each of the eight samples by taking 500 mL of distilled water as the dissolution medium. The conditions were 50 rpm paddle rotation speed and 37 ± 0.5 °C temperature. Then, approximately 160 mg of the samples were weighed and made up to 10 mL using distilled water in separate falcon tubes. The percentage transmittance of the samples was observed at 638.2 nm wavelength, using a UV visible spectrophotometer.

Two different co-surfactants were chosen for the co-surfactant screening procedure, namely Transcutol HP and Propylene Glycol, respectively. An RPC equivalent to 50 mg resveratrol was dissolved in 300 mg of the selected oil; then, 200 mg of the selected surfactant and 100 mg of both the co-surfactants were added to each vial before dispersing them completely. The screening procedure was the same as that of the surfactants.

##### Pseudo-Ternary Phase Diagram Study

The concentration of oil, surfactant, co-surfactant, and TPGS was determined by constructing pseudo-Ternary Phase Diagrams (PTPD), using the Prosim Ternary Diagram software (ProSim^®^ Inc., Philadelphia, PA, USA). The 10–40%, 60–90%, and 10–40% weight ratios of oil, surfactant, and co-surfactant, respectively, were observed in the phase diagram study, which comprised 11 points. For all the regions studied, emulsification time and percentage transmittance were recorded. The region with the shortest emulsification time and the highest percentage transmittance was selected for the further optimization process.

##### Determination of TPGS and RPC Concentration 

The selected region of PTPD was used to determine the appropriate concentration of Vitamin E TPGS. A total of 50 mg and 100 mg of TPGS in 1 g of the total mixture were examined within the previously selected region. The region with the shortest emulsification time and the highest percentage transmittance was selected for the further optimization process. Four different concentrations of RPC (1%, 1.5%, 2%, and 3%) were examined in the same PTPD region with the same selection criteria.

#### 3.2.4. Preparation Procedure

The RPC and TPGS were dispersed in the co-surfactant at 50 °C using a thermometer and a water bath. Then, oil, surfactant, and TPGS were added with vortex for the required amount of time. 

#### 3.2.5. Design of Experiment (DoE)

The mathematical optimization of the RPC-SNEDDS was carried out using Design Expert^®^ Software (Stat-Ease, Version 13, Minneapolis, MN, USA). The region attained from the PTPD investigation was utilized as the constraint for independent variables of mixture design: 10% ≤ Labrafil^®^ M 1944CS ≤ 30%, 40% ≤ Kolliphor^®^ RH 40 ≤ 60%, and 30% ≤ Transcutol^®^ HP ≤ 40%. In addition, four CQAs were served as dependent variables, ET (s, Y_ET_), GS (nm, Y_GS_), PDI (Y_PDI_), and Release (%, Y_Release_). Each CQA response was analyzed by following polynomial equations proposed by Goos et al., which provide the best-fitted polynomial model and determine the effect of each component on the response [30].
$$E(Y)=\sum_{i=1}^{q}\beta_{i}x_{i}+\sum_{i=0}^{q-1}\sum_{j=i+1}^{q}\beta_{ij}x_{i}x_{j}+\sum_{i=0}^{q-1}\sum_{j=i+1}^{q}\gamma_{ij}x_{i}x_{j}\;(x_{i}-x_{j})+\sum_{i=1}^{q-2}\sum_{j=i+1}^{q-1}\sum_{k=j+1}^{q}\beta_{ij}x_{i}x_{j}x_{k}$$

##### Generation and Verification of Design Space

The design space was generated using graphical optimization where CQA targets served as the optimization goals, along with 95% prediction intervals. The desirability function was employed to find an optimal formulation exhibiting the highest desirability value. The design space was externally and internally verified using an independent data set comprised of eight verification runs, following our previously developed method with a slight modification [12]. The optimal design point (VR1, desirability = 0.728) was also executed for verification purposes. Three design points were selected from inside the design space in which two points were axial check blends and the third was on the border of the design space. Two design points were picked from the conservative region of the design space. Two axial check blends were selected from outside the design space for the verification run. The predicted mean of the CQA responses derived from those verification runs along with their lower limit and higher limit of 95% prediction intervals were compared against their correspondent experimental value. 

### 3.3. In Vivo Pharmacokinetic Study

In vivo pharmacokinetic study was carried out to investigate the extent of bioavailability enhancement from the optimized SEDDS. The approval of in vivo study was obtained from the Animal Ethics Committee of Bangladesh Atomic Energy Center, and the guidelines were followed throughout this study. Male Wister Albino rats weighing 250–280 g were used for this experiment, and each group contained six rats (*n* = 6). One hourprior to oral dosing, Group-III received 3 mg/kg cycloheximide by intraperitoneal injection with saline to block the lymphatic drug transport. The rest of the groups were pretreated with the same volume of saline only. Group-I received pure HPMC suspension containing 25 mg resveratrol by oral gavage. Group-II received resveratrol–phospholipid complex containing 25 mg resveratrol. Group-III and Group-IV received the optimal SEDDS formulation containing 25 mg resveratrol. The blood samples were collected from the tail vein at 0, 5, 30, 60, 120, 180, 240, 300, 360, 420, and 720 min after oral gavage. The blood was centrifuged at 10000 rpm for 15 min, and the plasma was collected from the supernatant. The plasma was stored at −20 °C until further analysis.

#### Sample Preparation

The sample preparation procedure was slightly modified as described by Kuk et al. [31]. In a centrifuge tube, 50 L of PBS (30 mM, pH 6) was obtained and combined with the plasma sample. Then, 300 µL of ethyl acetate was added and vortexed for 30 s. The resulting mixture was centrifuged at 10,000 rpm for 10 min, and the upper organic layer was carefully transferred to another centrifuge tube. The extraction process with ethyl acetate was repeated twice more. The organic layers were then evaporated at 35 °C in the presence of nitrogen gas. Following that, the residue was reconstituted with 500 µL of methanol, followed by 10 min of centrifugation at 15,000 rpm. Finally, the supernatant was then transferred to the sample vial for HPLC analysis. 

The pharmacokinetic parameters were analyzed using PKSolver, a free Add-ins program of Microsoft Excel (Microsoft Office 2019). The observations of pharmacokinetic parameters were expressed as mean ± standard deviation (SD). The percentage of drug transported through the lymphatic pathway was calculated using the method adopted by Patel et al. [32].
(1)$$\text{Lymphatic transport}\;(\%)=\;\frac{\mathrm{AUC}\;\mathrm{of}\;\mathrm{saline}\;\mathrm{treated}\;\mathrm{group}-\mathrm{AUC}\;\mathrm{of}\;\mathrm{cycloheximide}\;\mathrm{treated}\;\mathrm{group}}{\mathrm{AUC}\;\mathrm{of}\;\mathrm{saline}-\mathrm{treated}\;\mathrm{group}}\times 100$$

### 3.4. HPLC Method

The concentrations of resveratrol in the samples were determined by the previously developed method using Dionex^TM^/Thermo^TM^ UltiMate 3000 HPLC System (LPG-3400 SD pump, DAD-3000 detector) with an Acclaim 120 C18 column (150 mm × 4.6 mm, 5µm, 120 Å) [33]. The mobile phase composition was 30 mM of phosphate buffer with a pH of 4.85 and acetonitrile at a 30:70 *v/v* ratio. The flow rate of the mobile phase was 1 ml/min; the injection volume was 2 μL; and the detection wavelength was 310 nm. The calibration curve was prepared using the 0.005 µg/mL to 8 µg/mL concentration range with R^2^ value of 0.9993.

### 3.5. Characterization of SEDDS

#### 3.5.1. Emulsification Time (ET)

The emulsification time was determined using the USP dissolution apparatus II (Copley Scientific, Model No. NE4-COPD, Nottingham, UK). Briefly, 1 g of the formulation was added to the 500 mL of distilled water at 37 °C under the rotation speed of 50 rpm. The time required to disperse the formulation entirely was determined as the emulsification time.

#### 3.5.2. Globule Size (GS) and Polydispersity Index (PDI)

The GS and PDI were measured by the dynamic laser light scattering (DLS) technique by Malvern Zeta Sizer NanoZS (Malvern Instruments, Malvern, UK). The formulation was diluted 100 times with double distilled water, and an adequate volume of the diluted sample was taken for the analysis.

#### 3.5.3. Transmittance Percentage (TP)

The formulation was diluted 100 times with double distilled water to measure the transmittance percentage of the formulation. The transmittance percentage of the diluted samples was determined by a UV-visible spectrophotometer (Shimadzu Corporation, Model No. UV-VIS 1600, Kyoto, Japan) at 638.2 nm. 

#### 3.5.4. In Vitro Drug Release

The in vitro drug release was performed as described by Amelia M. Avachat [34]. As the release media, 900 mL of simulated gastric fluid (SGF) was used for the initial 2 h and after that simulated intestinal fluid (SIF) was used. A total of 500 mg sample of each formulation with 1.5 mL SGF was loaded inside the overnight soaked dialysis bags (MWCO 12–14 kDa, HIMedia, Mumbai, India). The dialysis bags were then tied to the paddles of the dissolution apparatus II using thread. The paddles were lowered, and the apparatus was operated at 50 rpm and 37 ± 0.5 °C. A 5 mL sample was withdrawn from SGF at 5 min, 15 min, 30 min, 45 min, 1 h, 1.5 h, and 2 h, then the dialysis bag was transferred to the SIF. The same volume of sample was withdrawn from SIF at 2.5 h, 3 h, 3.5 h, 4 h, 5 h, 6 h, 7 h, and 8 h times. After sample withdrawal, it was subsequently replaced with 5 mL of SGF or SIF to maintain the sink conditions. Next, the samples’ absorbance was recorded in the UV-Visible spectrophotometer at 306 nm wavelengths.

## 4. Result and Discussion

### 4.1. Characterization of RPC

#### 4.1.1. FTIR, RD and DSC 

The resveratrol and phosphatidylcholine showed characteristic –OH stretch at 3550–3200 cm^−1^ area in Figure 1a. However, upon successful complexation of resveratrol with phosphatidylcholine at a 1:1.2 stoichiometric ratio, this stretch was not observed. The spectra were consistent with that of the previously obtained reports [35]. The XRD pattern of complexes and their constituents were determined (Figure 1b) to confirm the complexation between resveratrol and phosphatidylcholine. Pure resveratrol showed the characteristic peaks at 6.66, 16.3, 19.7, 22.52, 23.66, 25.24, and 28.3, demonstrating the crystalline form of resveratrol [36]. A large (wide) diffraction peak was observed for phosphatidylcholine, indicating the amorphous structure of phosphatidylcholine. Three intense peaks of resveratrol at 19.72, 22.52, and 28.3 still showed in the physical mixture of resveratrol and phosphatidylcholine. However, no typical crystalline peak of resveratrol was observed in the resveratrol–phospholipid complex; instead, it showed a broad-amorphous peak similar to that of phosphatidylcholine. It demonstrates the dropping of resveratrol’s crystal property completely and the successful formation of the resveratrol–phospholipid complex. The DSC thermogram revealed that the pure resveratrol had an intense exothermic peak at 267 °C (Figure 1c). The physical mixture of resveratrol and phosphatidylcholine also showed this exotherm however with less intensity. The exothermic peak was totally absent in the resveratrol–phospholipid complex, indicating that the complex was formed between resveratrol and phosphatidylcholine. The surface morphology of resveratrol–phospholipid complex was observed by SEM (Figure 1d). No crystalline-type structure was seen, indicating the absence of drug in crystalline form in resveratrol–phospholipid complex.

#### 4.1.2. Solubility Studies

Solubility analysis revealed differences in RPC’s oil and water solubility, compared to pure resveratrol. The pure resveratrol showed water solubility of 0.028 ± 0.017 n-octanol solubility of 0.136 ± 0.026. The water solubility of resveratrol after complexation with phospholipid was determined to be 0.129, which was 4.61-fold more than that of pure resveratrol. The improvement of resveratrol’s water solubility improves the dissolution profile of resveratrol in the gastrointestinal tract (GIT) [13]. The n-octanol solubility of resveratrol was found to be 0.2856, which was 2.1-fold higher than pure resveratrol. The increased oil solubility of payloads enhances permeability and the possibility of lymphatic drug transfer [21].

### 4.2. Determining QTTPs and CQAs

The QbD workflow began with defining QTPP terms and their targets with the appropriate justification in Table 1 based on the quality, safety, and efficacy of the intended drug product to be developed. The QTPPs elements were categorized as MUSTs, NEEDs, and WANTs, which were expressed as high-, medium-, and low-risk QTPPs. The high-, medium-, and low-risk QTPPs were also graded on the RPN scale of 9, 3 and 1, respectively, depending on their clinical importance, patient, and product requirement. The QTPPs were represented as H, M and L, representing high-, medium-, and low-risk, respectively. The QAs with their appropriate target and justification were also described in Table 2.

### 4.3. Risk Assessment

To identify the critical parameters, the RAM of QTPP-QA was determined (Figure 2a) using the previously mentioned risk scale (0–9). The interdependence between QTPPs and QAs is decided throughout the RAM generation process: how much is each QA related to each QTPP? The final risk score of each QA was obtained from the RAM analysis, which ranged from 27 to 177. QAs with a risk score of 1–60, 61–120, and 120–180 were classified as low-, medium-, and high-risk QAs, respectively. Release, ET, GS, and PDI were categorized as high-risk and medium-risk QAs as per this classification and designated as CQAs. Although transmittance percentage was not classified as a CQA, it will be employed in conjunction with ET to characterize the self-forming emulsion prior to the DoE investigation. From the optimization process, establishing design space, and design space verification study, the globule size of the emulsion will be characterized by the dynamic light scattering system (DLS). Assay and content uniformity is an important QA that needs to be characterized during the drug product development process. Since SEDDS is a preconcentrate product and payload is solubilized in it, assay and content uniformity variability is less critical and less interdependent with QTPPs [39]. Therefore, it resulted in a low-risk CQA.

The initial risk analysis for MAs/PPs to determine CMAs/CPPs were performed in two steps. In the first step, a diagram (Appendix A) was generated incorporating the factors that may affect QAs, including prospective MAs and PPs, testing methodology, equipment, and any other factors that might impact the QAs. Apart from prospective MAs and PPs, careful considerations were taken for those factors affecting QAs throughout this product developmental process. In the second step, important MAs/PPs were picked from the prospective MAs/PPs of the diagram to generate RAM of the QA-MA/PP relationship (Figure 2b). The interdependency rating between each MA/PP and each QA as risk-level was determined based on the following consideration/s; how much an MA/PP can affect a QA, and/or if an MA/PP goes out of specification, then it can affect a QA at what magnitude. A QA-MA/PP relationship was considered low-risk (=1) when an MA/PP has little or no possibility of impacting a QA. The impact could be significant but manageable and the impact could be significant but non-manageable were regarded as medium-risk (=3) and high-risk (=9), respectively. 

From the RAM of QA-MA/PP, the final risk score of each MA/PP was obtained, ranging from 1.72 to 7.64 (Figure 2e). Based on the risk score, MAs/PPs were classified as low-risk, medium-risk, and high-risk, when the final risk score were 1–3, 3.01–6, and 6.01–9, respectively. Following this classification, the surfactant concentration, surfactant type, oil type, oil concentration, surfactant type, co-surfactant concentration, and TPGS concentration were regarded as the high-risk MAs/PPs. Complex concentration was classified as medium-risk MA/PP, and other MAs/PPs were classified as low-risk MAs/PPs. The high-risk and medium-risk MAs/PPs were designated as CMAs/CPPs.

### 4.4. Risk Reduction of MAs by Experimental Approaches

The severity risk scores of MAs were reduced by their appropriate section or concentration range determination. The risk scores of oil, surfactant, and co-surfactant types were reduced by their appropriate type or favorable grade for lymphatic drug transport. The risk scores of oil concentration, emulsifying agent concentration, co-surfactant concentration, and TPGS concentration were reduced by determining their concentration or concentration range to meet the appropriate CQA target.

#### 4.4.1. Oil Phase

The solubility of the RPC in six oils was determined, and the extent of solubility is shown in Figure 3a. These oils were comprised of Long-Chain Fatty Acids (LCFAs), their mono-, di-, and triglycerides, or their semisynthetic derivatives. The oil phase from long-chain derivatives was chosen due to their superior capacity to assist lymphatic drug transport than that of medium-chain derivatives [8]. The results showed that the complex is most soluble in Labrafil^®^ M 1944 CS (94.57 mg/g) and it was therefore selected as the oil phase.

#### 4.4.2. Surfactant and Co-Surfactant

Upon dilution in an aqueous media, the surfactant and the co-surfactant of SEDDS facilitate the emulsification of the drug-loaded oil phase. Higher solubilization capacity of surfactant and co-surfactant may increase the payloads in SEDDS preconcentrate only. However, it cannot ensure a high payload in the oil phase after the self-emulsification process because during the self-emulsification process, a notable portion of surfactant and co-surfactant could be distributed to the aqueous phase. Furthermore, after the self-emulsification process, a small portion of surfactant and co-surfactant could be released from the oil phase to the aqueous phase, leading to the precipitation of payloads [40,41]. The aim of selecting a surfactant and co-surfactant was to emulsify the oil phase. Therefore, during the selection of surfactant and co-surfactant, the self-emulsification capacity of the oil phase was considered rather than the drug-solubilization capacity. 

The emulsification capacities (ET and TP) of Kolliphor^®^ RH40, Kolliphor EL, Gelucire 44/14, Gelucire 48/16, Gelucire 50/13, Labrasol ALF, Tween 80, and Tween 20 with Labrafil^®^ M 1944 CS are shown in Figure 3b. Kolliphor^®^ RH 40 showed the highest percentage transmittance (77.20%) and the lowest emulsification time (1.5 min) compared to other surfactants, and it was selected as the emulsifying agent. The addition of a co-surfactant to the surfactant may enhance stability, interfacial fluidity, and homogeneity of self-formed nanoemulsion [42]. Therefore, the co-surfactant was selected based on the capacity to aid surfactant during the self-emulsification process, i.e., reduce the ET and increase the TP of Labrafil^®^ M 1944 CS and Kolliphor^®^ RH40 mixture. Transcutol^®^ HP, with Labrafil^®^ M 1944 CS and Kolliphor^®^ RH 40 mixture, offered the shortest emulsification time and the highest percentage compared to the propylene glycol (Figure 3c). Thus, Transcutol^®^ HP was chosen as the co-surfactant. 

#### 4.4.3. Determining the Concentration Range of Oil, Surfactant, and Co-Surfactant

Selective regions were studied within the PTPD with the concentration of oil of 10–30%. Since above this limit, the oil causes an increase in mean droplet size and PDI as per the study published by Ma et al. [43]. The surfactant concentration studied was 30–70%. The co-surfactant concentration was 10–40% since up to a specific concentration of Transcutol HP (depending on the type of oil and type of surfactants used), the mixture forms small droplets, and above that, the droplet size increases drastically according to Xi et al. [44]. Therefore, further increasing the amount of Transcutol HP would reduce the number of excipients that aid in bioavailability enhancement. Therefore, to understand the effect more closely, 10–40% limits of Transcutol HP were considered, and later on the best points were selected.

The phase diagram showed the 10–30%, 30–70%, and 10–40% weight ratios for Labrafil^®^ M 1944 CS, Kolliphor^®^ RH40, and Transcutol^®^ HP, respectively, and that comprised 11 points, as depicted in Figure 4a. The weight ratio of these points with their corresponding ET and TP are shown in Appendix A. The TP ≥ 90% and ET ≤ 5 min were set as the margin initially to select the point of PTPD. The waiver on ET was executed initially, as TPGS addition may reduce the ET to the desirable limit. Within these criteria, 2–8 points were selected (Figure 4b), and the rest of the points were exempted. 

#### 4.4.4. Determining the Optimal Concentration of TPGS and RPC

TPGS addition to preconcentrate in 2–8 points at 5% and 10% of total weight and their corresponding ET and TP are listed in Appendix A. Although 5% TPGS addition to the preconcentrate reduced the ET, but not in 60 s for most points. Besides, the fall of TP by less than 90% was observed for all points. The addition of 10% TPGS did not significantly change TP, however it reduced the ET in ≤ 60 s for five points. Therefore, 10% TPGS was selected for further study within the region comprising five points (Figure 4c) and chosen as the optimal region for RPC concentration investigation. It revealed that though TPGS was mainly added to facilitate intestinal lymphatic transport of resveratrol, it also exhibited emulsification efficiency. Besides promoting lymphatic drug transport, Valicherla et al. also utilized TPGS as an emulsifying agent along with Gelucire 44/14, resulting in an increase in emulsification efficiency [45]. A total of 1% (10 mg) resveratrol equivalent RPC was determined as the optimum payload concentration compared to other concentrations examined on a trial-and-error basis (data not shown), as it did not affect the emulsification efficiency notably of the TPGS optimized region.

Following the experimental approach of risk reduction as discussed above, the final risk scores of MAs/PPs obtained from the further RAM analysis were shown in Figure 2e alongside the initial risk scores of MAs/PPs. MAs/PPs with a score of 1 were low-risk MAs/PPs and waived from further investigation. The risk scores of oil concentration, surfactant concentration, and co-surfactant concentration were all 4.29, indicating medium-risk MAs/PPs and determined as CQAs according to the predefined risk scale. Therefore, an appropriate DoE was required to further optimize the concentration range of 3 MAs along with the predetermined optimal concentration of TPGS and RPC loading 433.

### 4.5. Mixer Design (DoE)

Mixer design was carried out to define a design space, find an optimal formulation, and elucidate the effect of each mixture component on each CQA response. As three CMAs were the concentration of oil, surfactant, and co-surfactant, choosing a mixer design was more appropriate than other factorial designs (DoE) [46]. As independent variables, the oil, surfactant, and co-surfactant concentration were expressed as X_1_, X_2_, and X_3_, respectively. The CQA responses, such as emulsification time, globule size, PDI, and Release, were expressed as Y_ET_, Y_GS_, Y_PDI_, and Y_Release,_ which served as the dependable variables. A 13-run comprising mixer design was summarized in Table 3. The best-fitted polynomial model for each CQA response was selected, which exhibited a highly significant sequential *p*-value, the high regression R^2^, predicted R^2^, and experimental R^2^ with the lowest difference between predicted and obtained R^2^. The fitness summary of the polynomial model for each CQA is listed in Appendix A. The co-efficient values of X_1_, X_2_, and X_3_ were correlated statistically with responses, and the *p*-value was less than 0.05 in all cases. 

#### 4.5.1. Emulsification Time (Y_ET_)


$$Y_{\mathrm{ET}}=25.64\;X_{1}+64.08\;X_{2}+76.24\;X_{3}-0.1893\;X_{1}X_{2}-63.04\;X_{1}X_{3}-228.39\;X_{2}X_{3}-283.84\;{X_{1}}^{2}X_{2}X_{3}+1848.82\;X_{1}X_{2}^{2}X_{3}-720.48\;X_{1}X_{2}X_{3}^{2}$$


From the above polynomial model, it has been found that Labrafil^®^ M 1944 CS, Kolliphor^®^ RH 40, and Transcutol HP increased the emulsification time individually (Figure 5a and Table 3). However, when used in combination, Labrafil^®^ M 1944 CS and Transcutol HP showed a decrease in emulsification time, however, the effect was profound when Transcutol HP and Kolliphor^®^ RH 40 were used. Transcutol HP and both Transcutol HP and Kolliphor^®^ RH 40 have surface-active properties and work synergistically to reduce surface tension. On the other hand, when Labrafil^®^ M 1944 CS was combined with Kolliphor^®^ RH 40, the reduction in emulsification time was less significant. This was probably due to the effect of oil molecules on Kolliphor^®^ RH 40’s ability to reduce the interfacial tension [47]. When the three were used in combination, the increasing concentrations of Labrafil^®^ M 1944 CS or Transcutol HP decreased the emulsification time. However, the increasing concentration of Kolliphor RH 40 augmented the emulsification time the most, which may be due to its large molecular size and high viscosity.

#### 4.5.2. Globule Size (Y_GS_)

**Y**_**GS**_ = + 24.89 X_1_ + 37.67 X_2_ + 446.45 X_3_ − 33.23 X_1_X_2_ − 857.30 X_1_X_3_ − 879.78 X_2_X_3_ + 1263.37 X_1_X_2_X_3_ + 88.75 X_1_X_2_(X_1_ −X_2_) + 554.90 X_1_X_3_(X_1_−X_3_) + 572.29 X_2_X_3_(X_2_−X_3_)

The polynomial equation shows that the droplet size was increased by all the components individually (Figure 5b). However, the increasing Kolliphor^®^ RH 40 concentration significantly increased the droplet size and it was more than that with Labrafil^®^ M 1944 CS (Table 3). However, the latter followed a cubic pattern, with increasing droplet size until a certain point was reached, after which the size was reduced. A similar observation was found by Ma et al. [43]. The most significant reduction in droplet size was caused when Kolliphor^®^ RH 40 was increased together with Transcutol HP because of their surface-active properties. A similar response was observed when the concentration of Labrafil^®^ M 1944 CS was increased proportionately with Transcutol HP. Furthermore, droplet size reduction efficiency followed the pattern, Labrafil^®^ M 1944 CS < Kolliphor^®^ RH 40 < Transcutol HP, where Labrafil^®^ M 1944 CS caused the least droplet size reduction. It might be due to the formation of a film surrounding the droplets by oil molecules, which entrapped the surfactants, reduced their surface-active properties, and acted as a ‘stabilizer’ according to Malkani. et al. [48]. Transcutol HP reduced droplet size extensively until a certain point, after which the droplet size increased again.

#### 4.5.3. PDI (Y_PDI_)

**Y_PDI_** = 0.1894 X_1_ + 0.6954 X_2_ + 10.11 X_3_ − 0.6684 X_1_X_2_ − 19.83 X_1_X_3_ − 20.61 X_2_X_3_ + 20.93 X_1_X_2_X_3_ + 1.90 X_1_X_2_ (X_1_−X_2_) + 12.83 X_1_X_3_(X_1_−X_3_) + 14.20 X_2_X_3_(X_2_−X_3_)

According to the equation and contour plot (Figure 4c), the PDI value increased with each component (Table 3). However, the PDI decreased when at least two of the three components were used together. However, the PDI value increased as more Kolliphor^®^ RH 40 was used and decreased when the Transcutol HP increased, possibly due to the latter’s strong surface-active properties that thoroughly homogenize the mixture. However, Labrafil^®^ M 1944 CS did not significantly change the PDI value when its concentration was increased. The possible explanation could be that Kolliphor^®^ RH 40 increases the ‘Ostwald Ripening Rate’ where the smaller particles contract and the larger particles become more significant, increasing the PDI. This phenomenon is further explained by Lia Zeng [49].

#### 4.5.4. Release (Y_Release_)

**Y_Release_** = 61.71 X_1_ + 78.78 X_2_ − 299.20 X_3_ + 64.22 X_1_X_2_ + 817.51 X_1_X_3_ + 802.13 X_2_X_3_ − 1153.47 X_1_X_2_X_3_ + 44.95 X_1_X_2_ (X_1_−X_2_) − 554.43 X_1_X_3_(X_1_−X_3_) − 557.28 X_2_X_3_(X_2_−X_3_)

The polynomial equation and contour plot (Figure 5d) confirmed that the formulations rich in Labrafil M 1944 CS had the least drug release in 24 h (Table 4). On the other hand, the percentage of drug release from formulations rich in Transcutol HP was significantly higher. However, since both follow a cubic type response, the drug release is comparatively lower in extreme concentrations of Transcutol HP. This data shows that drug release was related to droplet size in an inverse relationship; the percentage of drug release was increased when the droplet size was reduced. Furthermore, all the formulations in the experiment formed droplets in the range of 18–40 nm. They are easily permeable through the intestinal membranes through cellular pores through passive diffusion [50].

#### 4.5.5. Design Space and Optimal Formulation

The optimization criteria used to generate the design space construction are summarized in Appendix A, and the resultant overlay plot is depicted in Figure 5e. The gray region of the overlay plot was a characterization space that was nevertheless out of CQA specifications. The conservative gray-yellow region was outside the design space due to constraints by 95% prediction interval limits, despite meeting the optimization criteria. This conservative region provides the confidence and managing uncertainty of prediction from the design space [51]. The yellow shaded region was the design space, where the predicted value of CQA responses met the acceptance criteria. A design point exhibiting the highest desirability value of 7.28 was selected as the optimal formulation. The optimal formulation showed the ET of 21.67 s, GS of 22.07 nm, PDI of 0.148 nm, and Release of 83.93%. According to the specification of Grade-A SEDDS, the preconcentrate has to be dispersed within 60 s with the formation of a clear to the slightly bluish-colored self-formed emulsion, and the globule size needs to be <100 nm [37,38]. After dilution, the preconcentrate of optimal formulation formed a slightly blueish-colored transparent emulsion with a mean GS of 22.07 nm within 21.67 s, confirming as a Grade-A SEDDS.

##### Verification of the Design Space

The CQA responses were fitted to the higher order of the polynomial model; therefore, the risk of data overfitting or high prediction variation in design space exists, even using 95% prediction intervals. For this reason, the design space needs to be internally and externally verified before further implementation [52]. The independent verification data set are listed in Appendix A, comprising the component ratio of verification design point, predicted mean, corresponding observed experimental value, lower limits, and upper limits of the 95% confidence intervals. The observed responses of verification runs inside the design space, and the border of the design space met the acceptance criteria of CQAs, demonstrating the predictive validity of the polynomial models. Furthermore, all observed CQA responses were closer to their corresponding predicted values, and they were found between the lower and upper limits of the 95% prediction intervals, demonstrating the robustness of formulation variables inside the design space.

### 4.6. In Vivo Pharmacokinetic Study

The results of pharmacokinetic parameters are shown in Table 4. Among these, two main pharmacokinetic parameters, namely AUC_0–720min_ and C_max_, are discussed in detail. The plasma concentration–time profiles of four study groups are illustrated in Figure 6. The plasma concentration–time curve of pure resveratrol showed a rise of blood concentration up to 30 min, when C_max_ was recorded as 0.24 µg/mL, and subsequently a fall up to 60 min. The slight increase in drug concentration was noted at 120 min, which resulted in a small second peak, followed by steadily falling till 240 min. This second peak is adequately described by the double peak phenomena of enterohepatic recirculation of resveratrol [53]. Following liver metabolism, only a tiny fraction of resveratrol reaches the systemic circulation, while the bulk is eliminated in the bile as the glucuronide/sulfate conjugate, followed by release into the small intestine, serving as a bolus dose. They are reabsorbed and reach the liver along with parent drugs; therefore, more drugs reach the systemic circulation compared to the previous point, explaining the appearance of a second peak [18,54].

Following oral administration of RPC to rats, the plasma concentration of resveratrol was much higher than that of resveratrol suspension. The C_max_ was raised by 9.46 times and AUC_0–720min_ by 10.58 times. It can be explained by RPC’s increased aqueous and lipid solubility than pure resveratrol, which resulted in increased dissolution and permeability. Most importantly, phospholipase A2 hydrolyzes diacylglycerol from the drug-phospholipid complex in GIT, releasing fatty acid and forming a drug-monoacyl phosphatidylcholine complex. In the presence of free fatty acid, cholecystokinin stimulates the liver and gall bladder to discharge bile into the duodenum. The drug-monoacyl phosphatidylcholine and bile salts form mixed micelles and are transported to the systemic circulation via the mesenteric lymphatic uptake, explaining the increased bioavailability of resveratrol from RPC [55].

The optimal SEDDS exhibited a 48-fold and a 16-fold increase in AUC_0–720min_ and C_max_, respectively, when compared to those for pure resveratrol and a 4.54- and a 2.01-fold increase in AUC_0–720min_ and C_max,_ when compared to those for RPC. These remarkable increases in AUC_0–720min_ and C_max_ can be explained by the following phenomena: Firstly, the smaller globule size of the self-formed emulsion increases the surface area to interact with the enterocyte surface, hence increasing permeability across the intestinal membrane. Additionally, the hydrophilic outer shells of Kolliphor^®^ RH40 and TPGS may facilitate diffusion across the unstirred water layer, a substantial barrier before the aforementioned permeation process [56]. Secondly, since resveratrol is a P-gp substrate, P-gp inhibitors in the formulation, such as TPGS and Kolliphor^®^ RH 40, could inhibit P-gp efflux and enhance the absorption process [57]. Thirdly and most importantly, SEDDS containing lipid-based excipients, notably TPGS, potentially enhance chylomicron secretion and facilitates lymphatic drug transport, bypassing hepatic metabolism and substantially increasing bioavailability [25,45,54,58]. Additionally, RPC increased the bioavailability of resveratrol via the intestinal lymphatic pathway, as explained above. Thus, incorporating RPC as the payload into SEDDS was another contributing factor to a substantial increase in the bioavailability of resveratrol.

Among the three proposed mechanisms of intestinal lymphatic transport, the transcellular route is the main route of lipophilic drug absorption from lipid-based formulations such as SEDDS [20]. Upon absorption, they stimulate chylomicron production within the enterocyte and enter the lymphatic capillaries as the drug-incorporated chylomicron, bypassing the hepatic portal circulation [59]. Cycloheximide inhibits lymphatic transport mainly by blocking chylomicron flow as well as M cells associated with lymphatic transport. However, they do not affect other non-lymphatic or hepatic portal absorption pathways [20]. Rats treated with cycloheximide + SEDDS showed a drastic drop in AUC_0–720min_ and C_max_, which were 8.95 and 4.42 times lower, respectively, compared to those treated with SEDDS alone. The cycloheximide-induced lymphatic transport inhibition of resveratrol could explain this drop in AUC_0–720min_ and C_max_ values. From Equation (1), it was revealed that approximately 91.14% of the total bioavailable drug was transported via the lymphatic pathway, whereas only 8.86% was transported via another route, preferentially the hepatic portal route.

The AUC_0–720min_ and C_max_ of cycloheximide + SEDDS treated group were lower than RPC and optimal SEDDS group; however, they were 5.37-fold and 4.29-fold higher than the pure resveratrol group. Therefore, apart from the lymphatic transport of resveratrol, the optimal SEDDS formulation showed the capacity of increasing 5.37-fold bioavailability by the hepatic portal route through some other mechanism. The nano-sized globule, increased solubility of SPC, permeation enhancing property, and P-gp inhibiting property of some excipients may be the contributing factors. The resveratrol is extensively glucuronidated by UGT1A1 and UGT1A9, whereas Labrafil^®^ M 1944CS and Kolliphor^®^ RH 40 have inhibitory effects on these enzymes. Yang et al. and Zhou et al. also showed the improved bioavailability of resveratrol utilizing glucuronidation inhibitory excipients in the formulation [14,60]. Thus, the inhibition of glucuronidation by Labrafil^®^ M 1944 CS and Kolliphor^®^ RH 40 might be another reason for increased bioavailability, despite restricting lymphatic transport.

Although the optimal SEDDS formulation remarkably enhanced bioavailability, the pharmacokinetic study was investigated in rats. A further pharmacokinetic study could be performed on healthy human subjects to observe bioavailability enhancement by optimal SEDDS in comparison with resveratrol suspension. Additionally, this study did not include an in vitro lipolysis test that could predict the fate of the SEDDS formulation upon digestion in GIT.

## 5. Conclusions

The formulation was designed and developed with the goal of targeting intestinal lymphatic drug transport to increase bioavailability of resveratrol. The optimal formulation was developed by selecting formulation components, determining their concentrations, and further optimizing the formulation to transport resveratrol through the lymphatic system, circumventing the hepatic portal pathway. The QbD-developed design space was highly robust since all CQAs responses of the verification data set were found within 95% prediction intervals. The optimized formulation exhibited Grade-A SEDDS, with a globule size of 22.07 nm and an emulsification time of 21.67 s. The pharmacokinetics study revealed that the optimal formulation increased bioavailability by 48 times compared to pure resveratrol, and the 91.14% drug was transported through the intestinal lymphatic pathway from the optimized SEDDS. This remarkable increase in bioavailability of resveratrol via the intestinal lymphatic route demonstrates the potential in combining the QbD approach to the lymphatic transport pathway to develop a better formulation. Since the extremely poor bioavailability of resveratrol has been a major barrier to extending its pharmacological effect to humans, the current study could be a breakthrough in overcoming these obstacles.

## Figures and Tables

**Figure 1 polymers-14-03220-f001:**
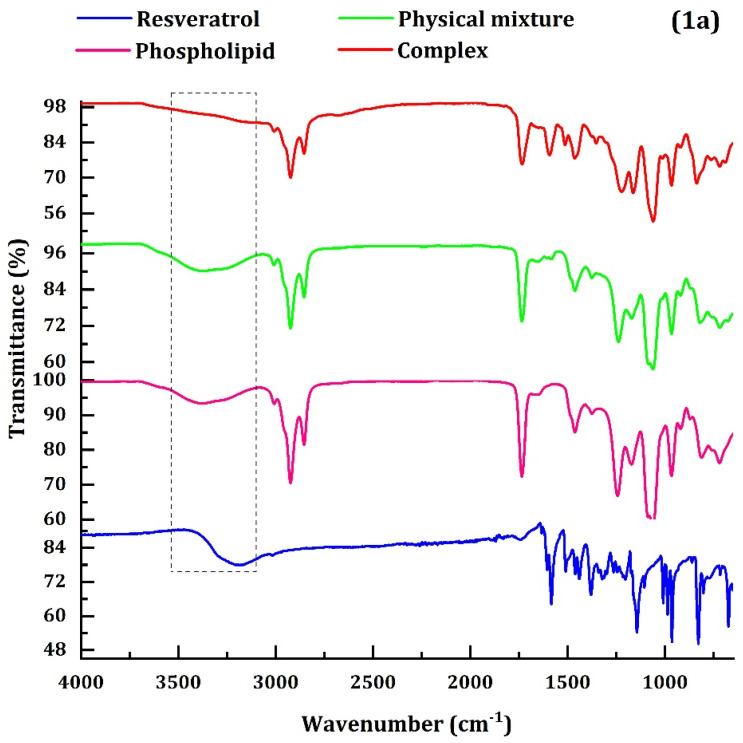
(**a**) FTIR spectroscopy and (**b**) X-ray diffraction patterns (**c**) DSC thermogram of pure resveratrol, phospholipid, physical mixture of resveratrol and phospholipid, resveratrol–phospholipid complex. (**d**) Scanning Electron Microscopic image of resveratrol–phospholipid complex.

**Figure 2 polymers-14-03220-f002:**
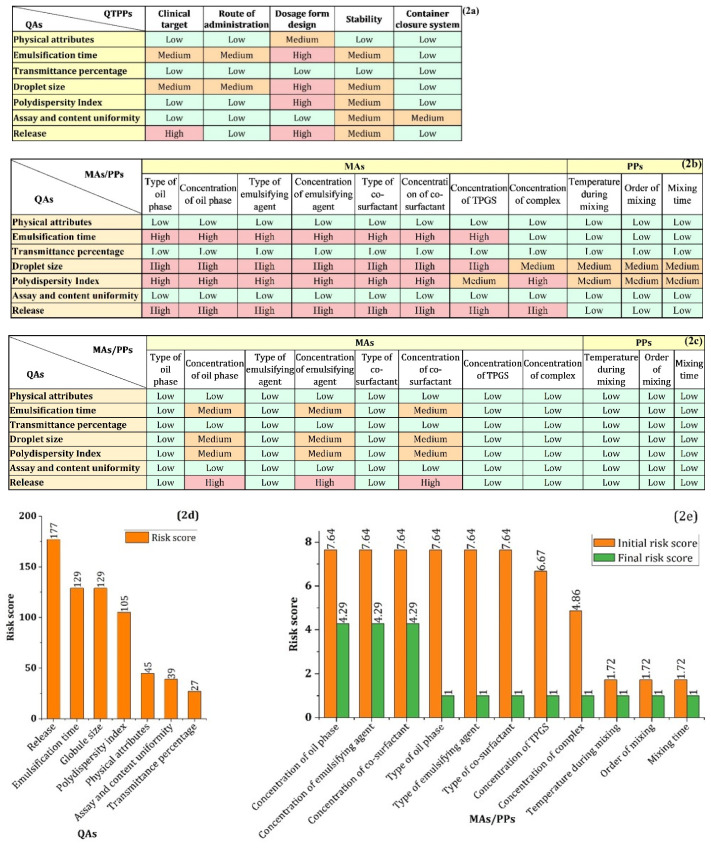
(**a**) RAM of QTPP-QA relationship, (**b**) RAM of QA-MA/PP relationship, (**c**) RAM of QA-MA/PP relationship after risk reduction, (**d**) risk score of QAs, and (**e**) risk score of MAs/PPs during initial risk analysis and after the risk reduction following the experimental approaches.

**Figure 3 polymers-14-03220-f003:**
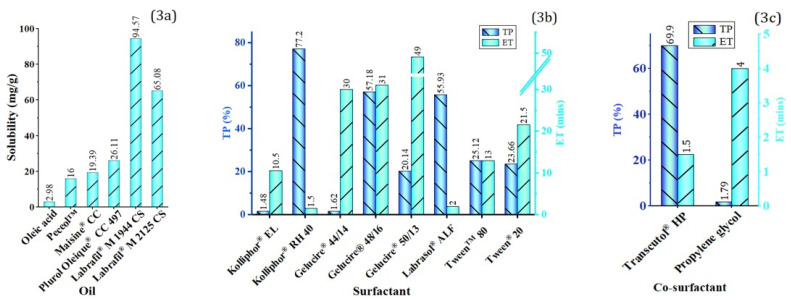
(**a**) RPC solubility in different oils, (**b**) ET and TP for Labrafil^®^ M 1944 CS with the different surfactant mixture, (**c**) ET and TP for Labrafil^®^ M 1944 CS and Kolliphor^®^ RH 40 with the different co-surfactant mixture.

**Figure 4 polymers-14-03220-f004:**
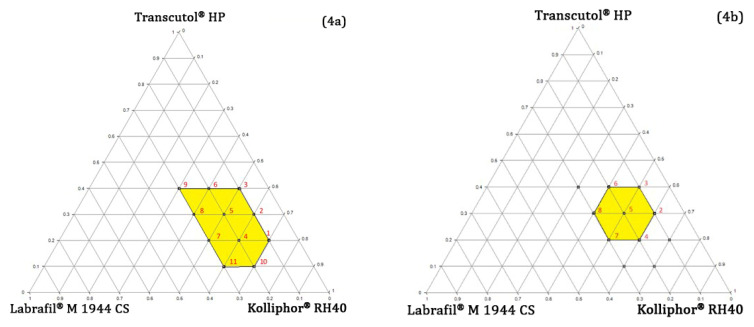
The ternary phase diagram indicates a suitable region for the oil, surfactant, and co-surfactant concentration.

**Figure 5 polymers-14-03220-f005:**
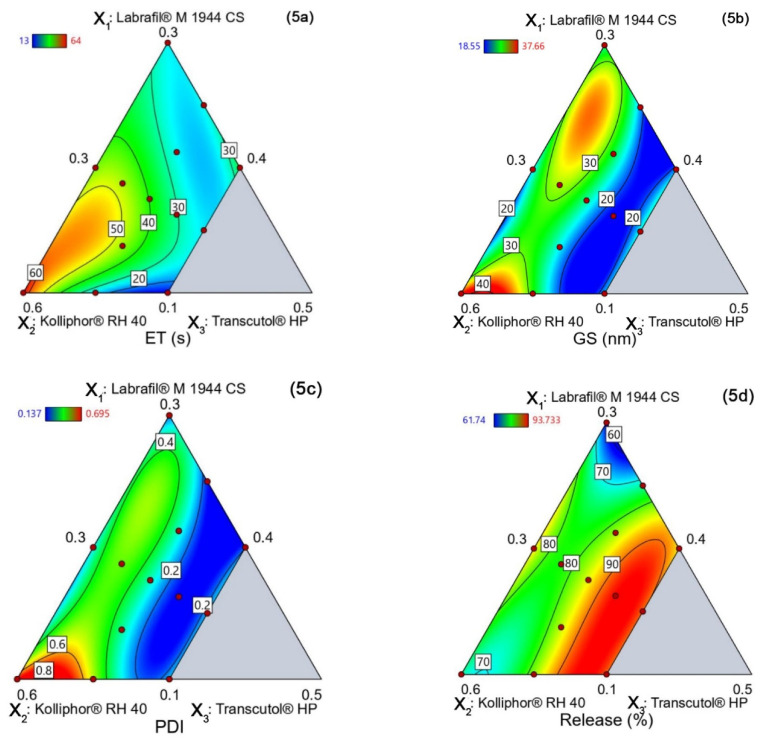
The contour plots of the four CQA responses (**a**) ET, (**b**) GS, (**c**) PDI, (**d**) Release, and (**e**) overlay plots represent the design space.

**Figure 6 polymers-14-03220-f006:**
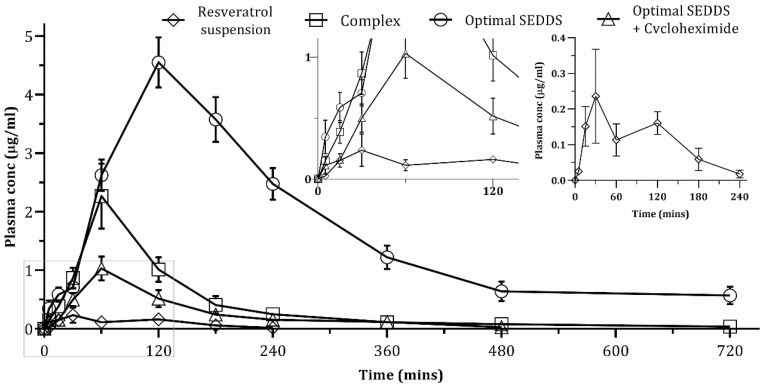
Plasma concentration–time profile of resveratrol in rats. Data are represented as mean ± SD (*n* = 6).

**Table 1 polymers-14-03220-t001:** QTPP elements with their targets and justification.

QTPPs	Target	Justification
Clinical target	Improving the bioavailability of resveratrol through lymphatic transport	Resveratrol undergoes extensive fast pass metabolism, and <1% becomes systemically bioavailable.
Route of Administration	Oral	The most convenient route of drug administration.
Dosage form design	Self-emulsifying drug delivery system	SEDDS offers higher drug loading and improved biopharmaceutical attributes of the loaded drug.
Stability	Six months (at least)	SEDDS is a preconcentrate dosage form that could be stored for a long time.
Container closure system	Amber glass container	Resveratrol undergoes photolytic degradation in the presence of light.

**Table 2 polymers-14-03220-t002:** QAs with their targets and justifications.

CQAs	Target	Justification
Physical attributes	No unpleasant color, odor, and taste	Those unpleasant attributes of formulation reduce the patient acceptability.
Transmittance percentage	≥90%	The transmittance percentage of ≥90% denotes ultrafine globules and is essential to maintain the class of Grade-A SEDDS, which can be used to characterize during the initial development of SEDDS instead of DLS [37].
Emulsification time	1–60 s	Rapid self-emulsion formation (within 60 s) is a requirement for Grade A self-emulsion [38].
Droplet size	10–50 nm	The globule size of ≤100 nm is the specification for Grade-A self-emulsion [37]. However, the globule size of <30 nm aid in permeation of the unstirred water layer and mucous layer.
Polydispersity index	0.2	The lower PDI values indicate a narrow globule size distribution and monodispersed globule.
Assay and content uniformity	100%	Assay and content uniformity are necessary to ensure the safety and efficacy of the drug product.
Release	80–100% at 8 h	A higher percentage of the drug needs to be released in the desired time.

**Table 3 polymers-14-03220-t003:** Mixer design comprising CMAs as dependable variables and CQAs as independent variables.

Run	Labrafil^®^ M 1944 CS (X_1_)	Kolliphor^®^ RH 40 (X_2_)	Transcutol^®^ HP (X_3_)	Emulsification Time, s (Y_ET_)	Globule Size, nm (Y_GS_)	PDI (Y_PDI_)	Release, % (Y_Release_)
1	0.1625	0.4625	0.375	34.31	18.55	0.137	93.73
2	0.25	0.4	0.35	26.12	21.79	0.157	72.82
3	0.2	0.5	0.3	44.97	22.89	0.274	86.24
4	0.175	0.475	0.35	41.17	25.77	0.301	86.02
5	0.1875	0.4875	0.325	47.21	31.69	0.476	78.39
6	0.2125	0.4375	0.35	23.89	26.08	0.3	82.18
7	0.1	0.5	0.4	13.21	22.08	0.249	90.43
8	0.15	0.45	0.4	22.87	23.875	0.2435	90.47
9	0.3	0.4	0.3	26.11	24.88	0.189	61.74
10	0.1	0.6	0.3	64.55	37.66	0.695	78.81
11	0.1	0.55	0.35	25.26	28.79	0.518	82.47
12	0.1375	0.5125	0.35	43.88	22.56	0.33	83.45
13	0.2	0.4	0.4	36.43	21.31	0.192	85.74

**Table 4 polymers-14-03220-t004:** Pharmacokinetic parameters of resveratrol in rat plasma. (Data represented as Mean ± SD, *n* = 6).

PK Parameters	Resveratrol Suspension	RPC	Optimal SEDDS	Optimal SEDDS + Cycloheximide
Area under curve, AUC_0–720min_ (µg /mL × min)	24.31 ± 4.31	257.15 ± 40.26	1167.39 ± 103.20	130.43 ± 21.14
Area under curve, AUC_0–∞_ (µg /mL × min)	25.31 ± 4.98	267.04 ± 41.62	1353.11 ± 170.97	134.37 ± 22.03
T_max_ (min)	30	60	120	60
C_max_ (µg/mL)	0.24 ± 0.12	2.27 ± 0.51	4.55 ± 0.39	1.03 ± 0.19
Plasma half-life, t_1/2_ (min)	35.5 ± 11.97	185.02 ± 44.78	217.26 ± 83.28	88.16 ± 16.3
Mean residence time, MRT (min)	100.75 ± 14.13	184.3 ± 15.95	357.33 ± 70.65	153.46 ± 21.33

## Data Availability

Not applicable.

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
