# Peer review of "A QbD Approach to Design and to Optimize the Self-Emulsifying Resveratrol–Phospholipid Complex to Enhance Drug Bioavailability through Lymphatic Transport"

_polymers, 2022, doi:10.3390/polym14153220_

Round 1
Reviewer 1 Report
Dear authors, it seems to me that you have done a good effort to perform the research presented in this manuscript. However, significant issues remain:
The abstract requires reconstruction. It is necessary to present it with a specific structure, rather than referring the results, because the reader is unaware of the methods you used.
The manuscript requires english language edition. There are so many mistakes which make it not easily readable.
A clear limitation paragraph is required just before the conclusion section.
The conclusions have to be re-written.
Author Response
Answer to the comments of Reviewer-1
- The abstract requires reconstruction. It is necessary to present it with a specific structure, rather than referring to the results, because the reader is unaware of the methods you used.
Answer: The abstract has been revised as per the recommendations.
- The manuscript requires english language edition. There are so many mistakes which make it not easily readable.
Answer: The manuscript has been critically edited.
- A clear limitation paragraph is required just before the conclusion section.
Answer: A limitation paragraph has been added as suggested.
- The conclusions have to be re-written.
Answer: We have rewritten the conclusion section.
Reviewer 2 Report
The manuscript entitled “Design and optimization of self-emulsifying resveratrol-phospholipid complex by QbD to enhance bioavailability through lymphatic transport'' contains some interesting findings, and it may ultimately be suitable for publication. However, the manuscript needs many more characterizations and data to improve the results and discussion. I thus recommend the paper be reconsidered after minor revisions.
The reviewer has the following comments
1. The title of the manuscript should be modified.
2. The abstract is tediously long, it should be terse.
3. TGA, DSC analysis, degradation, porosity, and zeta potential of the resveratrol-phospholipid complex should be reported
4. The morphologies (SEM), or other morphology images of the resveratrol-phospholipid complex should be reported.
5. The introduction section is very short and should be revised entirely so that the reader can clearly identify the scientific problems solved by this research. There are several biocompatible systems with a wide range of biomaterials, including hydrogels, nanoparticles, nanofibers, nanofilms, nanocomposites, etc., but why did the authors select only phospholipids? The authors should emphasize why the phospholipids are familiar, or favorable compared to other systems. Moreover, the information on biomaterials (Chitosan, Gelatin, Zein, and PCL) should be expanded in the introduction. Thus, the biomaterials research work from Narsimha Mamidi, Enrique V Barrera, and Alex Elias Zuñiga, should be quoted in the introduction, and other suitable sections of the manuscript.
It would be more realistic to cover such kind of biomaterials research work in the current manuscript. Which will enrich the quality of the current manuscript as well as inquisitiveness to the readers.
6. According to the revised data, the conclusions should be modified with more quantitative data.
Author Response
Answer to the comments of Reviewer-2
- The title of the manuscript should be modified.
Answer: The title of the manuscript has been modified as “QbD approach to design and optimize the self-emulsifying resveratrol-phospholipid complex to enhance drug bioavailability through lymphatic transport”
- The abstract is tediously long, it should be terse.
Answer: The length of abstract has been reduced as per the recommendation.
- TGA, DSC analysis, degradation, porosity, and zeta potential of the resveratrol-phospholipid complex should be reported.
Answer: We have included the results of DSC analysis of the resveratrol-phospholipid complex. However, the porosity measurement facility (by BET method) is not available in our country, so we are unable to produce this data. Since we have characterized the resveratrol-phospholipid complex by FTIR and DSC, as well as SEM, we think the resveratrol-phospholipid complex is well defined by these analyses.
- The morphologies (SEM), or other morphology images of the resveratrol-phospholipid complex should be reported.
Answer: The SEM image of the resveratrol-phospholipid complex has been incorporated in the revised manuscript.
- The introduction section is very short and should be revised entirely so that the reader can clearly identify the scientific problems solved by this research. There are several biocompatible systems with a wide range of biomaterials, including hydrogels, nanoparticles, nanofibers, nanofilms, nanocomposites, etc., but why did the authors select only phospholipids? The authors should emphasize why the phospholipids are familiar, or favorable compared to other systems. Moreover, the information on biomaterials (Chitosan, Gelatin, Zein, and PCL) should be expanded in the introduction. Thus, the biomaterials research work from Narsimha Mamidi, Enrique V Barrera, and Alex Elias Zuñiga, should be quoted in the introduction, and other suitable sections of the manuscript.
It would be more realistic to cover such kind of biomaterials research work in the current manuscript. Which will enrich the quality of the current manuscript as well as inquisitiveness to the readers.
Answer: The bioavailability of resveratrol is less than 1%. We undertook this study to increase the bioavailability of resveratrol by targetting the transportation of the drug through the intestinal lymphatic pathway. Therefore, we have selected the excipients which could facilitate the drug transportation via the lymphatic route, and these are described in the introduction section accordingly. Although other biomaterials like Chitosan, Gelatin, Zein, and PCL are well recognized in structuring the nanoparticles, they are rarely used in self-emulsifying drug delivery system (SEDDS) as these biomaterials have barely any role in facilitating the lymphatic drug transport. So, we excluded them from the introduction section.
However, TPGS is a well-recognized biomaterial and excipient which has a tremendous role in promoting lymphatic drug transport, and we utilized it in designing our SEDDS formulation. So, we have included a short description of TPGS in the revised version. We have also added the rationality of using phospholipid in the formulation in the revised manuscript.
- According to the revised data, the conclusions should be modified with more quantitative data.
Answer: With the incorporation of quantitative data, the conclusion section has been rewritten.
Round 2
Reviewer 1 Report
The authors answered to my requests/recommandations.